# Anatomical Tissue Engineering of the Anterior Cruciate Ligament Entheses

**DOI:** 10.3390/ijms24119745

**Published:** 2023-06-05

**Authors:** Clemens Gögele, Judith Hahn, Gundula Schulze-Tanzil

**Affiliations:** 1Institute of Anatomy and Cell Biology, Paracelsus Medical University, Nuremberg and Salzburg, Prof. Ernst Nathan Str. 1, 90419 Nuremberg, Germany; clemens.goegele@pmu.ac.at; 2Workgroup BioEngineering, Department Materials Engineering, Institute of Polymers Materials, Leibniz-Institut für Polymerforschung Dresden e.V. (IPF), Hohe Straße 6, 01069 Dresden, Germany; hahn-judith@ipfdd.de

**Keywords:** ACL, enthesis, ligament, synovioentheseal complex knee, tissue engineering, triphasic and graded scaffold, fibrocartilage, bone—ligament interface, zonality, tidemark

## Abstract

The firm integration of anterior cruciate ligament (ACL) grafts into bones remains the most demanding challenge in ACL reconstruction, since graft loosening means graft failure. For a functional-tissue-engineered ACL substitute to be realized in future, robust bone attachment sites (entheses) have to be re-established. The latter comprise four tissue compartments (ligament, non-calcified and calcified fibrocartilage, separated by the tidemark, bone) forming a histological and biomechanical gradient at the attachment interface between the ACL and bone. The ACL enthesis is surrounded by the synovium and exposed to the intra-articular micromilieu. This review will picture and explain the peculiarities of these synovioentheseal complexes at the femoral and tibial attachment sites based on published data. Using this, emerging tissue engineering (TE) strategies addressing them will be discussed. Several material composites (e.g., polycaprolactone and silk fibroin) and manufacturing techniques (e.g., three-dimensional-/bio-printing, electrospinning, braiding and embroidering) have been applied to create zonal cell carriers (bi- or triphasic scaffolds) mimicking the ACL enthesis tissue gradients with appropriate topological parameters for zones. Functionalized or bioactive materials (e.g., collagen, tricalcium phosphate, hydroxyapatite and bioactive glass (BG)) or growth factors (e.g., bone morphogenetic proteins [BMP]-2) have been integrated to achieve the zone-dependent differentiation of precursor cells. However, the ACL entheses comprise individual (loading history) asymmetric and polar histoarchitectures. They result from the unique biomechanical microenvironment of overlapping tensile, compressive and shear forces involved in enthesis formation, maturation and maintenance. This review should provide a road map of key parameters to be considered in future in ACL interface TE approaches.

## 1. Introduction

Due to their high biomechanical loading, ACL entheses are prone to injury. As reported previously, many ruptures of ACL and ACL grafts can be observed at the area of the femoral enthesis; in particular, fibers of the posterolateral ACL bundle have a high risk of rupture during pivot landings [1] (Figure 1). Typically, autografts or allografts are used for reconstruction. Hence, the bone attachment site represents a major point of weakness after ACL reconstruction [2], since most of the techniques do not properly restore the entheses [3]. The reasons for this include a poor spontaneous healing ability and the fact that the regenerative tissue in the bone attachment site is mostly composed of poorly organized Sharpey-like fibers, leading to the insufficient stability of the graft [4]. Sharpey fibers represent connective tissue fibers mainly consisting of collagen type I which attach to the periosteum, but do not reflect a gradual and biomechanically competent fibrocartilaginous enthesis. In addition, several studies have shown that impaired healing can lead to bone tunnel widening and a consequent laxity of the graft as synovial fluid infiltrates into the tunnel [5,6,7], thereby impairing integration. The consequences can include knee instability after ACL graft loosening [2] and even osteoarthritis (OA) [8]. The onset of OA, which is associated with an inflammatory microenvironment, might further interfere with the integration process.

The structural components of the enthesis act to minimize stress concentrations and control stretch at the interface [9]. To overcome auto- and allograft limitations in regard to the restricted availability of well-sized autografts and donor site morbidity, etc. [10,11], a tissue-engineered ACL substitute becomes a sizable vision [11]. In this regard, the idea of establishing an ACL on chip as a tool for future research towards a tissue-engineered ACL has been discussed previously in a review [12]. So far, tissue-engineered ACL grafts comprising the bone insertion site have been developed and tested in sheep and rabbit models [13,14,15]. The emerging research field addressing the reconstruction of the enthesis has been designated as interface TE. Patel et al. summarized current graft designs for rotator cuff tendon to bone and ACL ligament to bone interfaces [16]. However, groundbreaking published data describing a tissue-engineered ACL enthesis with a clear focus on modeling and mimicking the unique human ACL anatomy are scarce. Since the dimensions, fiber angle and composition of the enthesis are species-specific and unique at its particular localization, it is important to know more about the anatomical peculiarities of the ACL enthesis as a blueprint for future ACL enthesis TE.

## 2. Results and Discussion

### 2.1. Anatomy: The ACL Entheses

Since the enthesis acts as a functional unit allowing stress dissipation from the stretched ligament to its bone attachment sites, an organ concept arose: in the case of the enthesis of the Achilles tendon, it has been designated as an organ underlining the synergistic function of the different tissues (including the fat pad, fibrocartilage and synovial structures) forming and surrounding the enthesis [17]. This concept is transferable to the ACL entheses, where similar neighboring tissues (Hoffa fat pad, synovial fold and fibrocartilage) might contribute to their overall functions and homeostasis. An enthesis of the most common fibrocartilaginous type comprises four tissue compartments of increasing stiffness (ligament, non-calcified and calcified fibrocartilage, separated by the tidemark as a barrier between both and the underlying bone), generating a biomechanical gradient at the attachment to bone (Figure 2). The borders between the zones show undulating interdigitations [18], increasing the overall surface of contact areas and hence providing sufficient stability for repetitive loading. The tidemark is also a barrier between two nutrient diffusion fronts, from the subchondral bone and from the vascularized ACL midsubstance and outer synovial layer [19,20,21].

In case of the ACL, a femoral (attached to the bone of the inner medial site of the lateral femoral condyle) and a tibial (fixed at the anterior tibial plateau) enthesis exist (Figure 1). The anterior part of the ACL crosses over the anterior meniscotibial ligament (AMTL) of the lateral meniscus (Figure 1). Some fibers can blend with those of this AMTL [22]. Both entheses form angles between their underlying bones and the course of the midsubstance of the ACL. The immediate anatomical environment of the ACL comprises the Hoffa fat pad [23] and the layers of the joint capsules, since the ACL is localized between both layers (outer fibrous and inner synovial part) of the joint capsule (Figure 1) [24,25]. It is ensheathed by a synovial layer and often covered by a fold of the synovial membrane (*plica synovialis infrapatellaris*), which could contain fatty tissue and which contributes to nutrition of the ACL containing blood vessels. The blood supply for the ACL derives mostly from the middle genicular artery, which is released from the popliteal artery and penetrates posteriorly the joint capsule before its branches follow the synovial connective tissue to the ACL [22]. Like the complete midsubstance of the ACL, its entheses are also surrounded by the synovial membrane and hence are exposed to the intra-articular micromilieu, but the fibrocartilaginous parts of them do not receive blood supply. The latter changes in cases of knee injury and might also affect enthesis homeostasis [25]. The epiligament underneath the synovial membrane covering the ACL midsubstance continues into the periosteum at the bone attachment sites (Figure 2). Its cellularity differs from that in other epiligaments and hence has been attributed to the low healing capacity of the ACL compared with other ligaments [26]. The connective tissue at the intercondylar notch of the femur also covering the femoral enthesis could contribute to graft integration after ACL reconstruction and should be paid attention to [27].

### 2.2. Differences between the Femoral and Tibial ACL Entheses

The femoral and tibial entheses of the ACL differ in regard to their overall and zone dimensions, fibrocartilage amount and fiber insertion angles [28]. The femoral enthesis had a 3.9-fold significantly sharper attachment angle of the fibers compared with that of the tibial enthesis. It also showed a significantly larger (43%) calcified and significantly greater (226%) non-calcified fibrocartilage zone, particularly in the central attachment area. The authors reported that these differences might provide an explanation of why it is prone to injury, since there possibly arises a stress concentration at the inferior border of the ACL’s femoral enthesis [28]. The femoral enthesis has a rather semilunar shape or represents a segment of a circle [22], whereas the tibial exhibits rather an elliptic attachment area [29]. The dimensions were reported as 11–24 mm for the long diameter for the femoral attachment zone and 17 × 11 mm as diameters for the tibial one [22].

Moreover, another study showed that the two bundles of the ACL (AM: anteromedial and PL: posterolateral) displayed different insertion modes, whereby the AM bundle formed more intimate interdigitations with the subligamentous cortical bone compared to the PL bundle attachment site, suggesting structure–function relationships and underlining the importance of sub-bundle anatomy [30], which could be incorporated in ACL reconstruction strategies in future.

A cadaver study revealed that two millimeters from its direct femoral attachment, the ACL fiber bundles formed a flat ribbon in all investigated donors and both, the AM and PL bundles, were barely separated in this area. This ribbon followed continuously the posterior femoral bone cortex. It had a width of between 11.43 and 16.18 mm and was only 2.54–3.38 mm thick [31].

Suzuki et al. presented data concerning the thicknesses of the enthesis zones [32]. They found that the mean thicknesses of the uncalcified fibrocartilage (FC) at the femoral and tibial attachments ranged between 0.98 mm and 0.49 mm, respectively and the mean thicknesses of the calcified FC were 0.47 mm and 0.38 mm, respectively. The femoral and tibial attachment sites differed significantly in regard to their uncalcified FC zone thicknesses, but the calcified layer had a comparable thickness. The trabeculae of the subligamentous bone orientated mostly parallel to the ACL collagen fiber bundles at both enthesis, but this concerted fiber/trabeculae orientation was more pronounced at the tibial compared to the femoral side. Moreover, it was enhanced in the proximal posterior part of the femoral enthesis and in the anteromedial part of the tibial attachment, probably as a correlate of mechanical stress concentration in these areas. Hence, the authors concluded that the femoral enthesis compensates multidirectional stresses, whereas more or less unidirectional stress is received at the tibial insertion. Moreover, stress seems to be higher at both, the proximal posterior femoral enthesis part and the anteromedial area of the tibial attachment [32]. Not only morphological aspects can be distinguished as outlined above, but also the healing capacity differed at both ACL entheses as shown in a rabbit model: it was histologically superior in the femoral segment, whereas less graft remodeling and Sharpey’s fiber formation could be observed at the tibial side [33].

### 2.3. ACL and Enthesis Development

Healing and reconstruction often recapitulate stages of tissue development. Hence, it is important to understand the key developmental processes of the ACL and its entheses. The ACL becomes visible before joint cavitation and starts its development at the ventral side of the future knee joint, invaginating gradually into the intercondylar space of the joint but remaining in an extrasynovial position over the entire time until adulthood. It migrates posteriorly. Initially, it derives from the same blastoma as the menisci, suggesting that both share functional aspects [22]. After birth, maturational changes (increase in collagen content and mineralization, but lesser collagen fiber alignment) can be observed in the ACL enthesis as a result of loading and remodeling as shown in bovine samples [34].

The clonal expansion of Growth and Differentiation Factor (GDF)5-positive progenitors triggers enthesis formation [35]. As crucial developmental actors, Indian hedgehog (Ihh), ParaThyroid Hormone related protein (PTHrP) and Patched 1 (Ptc1) were localized in the developing enthesis. The latter are known to regulate chondrocyte proliferation and differentiation, being in a negative feedback interrelation. The transcription factors SOX9 and scleraxis regulating several key ECM gene expressions also seem to contribute to enthesis formation. The zone-specific expression of these factors might be responsible for graded tissue formation at the enthesis [36]. Before mineralization, collagen-type-I-producing cells become associated with collagen-type-II-synthesizing cells. As a starting point of mineralization, cells at the base of the enthesis start to express Ihh, alkaline phosphatase and collagen type X. Ihh-responsive glioma associated oncogene homologue (Gli)1-positive cells differentiate from non-mineralized ECM producing fibrochondrocytes in those synthesizing a calcifying ECM [35]. One could estimate that the final ECM expression pattern in the maturating enthesis is strongly regulated by mechanobiology [37].

### 2.4. Histoarchitecture, Cellular and Biochemical Pecularities

The ECM of the different enthesis zones shows a unique expression profile, summarized in Table 1. It comprises ECM components more typical of the dense regular connective tissue of the ACL (collagen types I and III) or more related to fibrocartilage (collagen type II, aggrecan), the latter to compensate pressure as well as components associated with calcification, e.g., collagen type X and hydroxyapatite. The orientation of the collagen fibers differs, e.g., more or less following one main direction (anisotrop, in response to unidirectional stress) or interwoven (isotrop) to compensate stress from different directions. The transition between zones is not abrupt, but forms a gradient [36]. The fibrocartilage of the enthesis generates a natural barrier separating the blood supply of the bone from that of the tendon/ligament [17].

### 2.5. Sex- and Training-Dependent Individual Aspects

It is well known that the anatomical dimensions of the knee joint differ substantially between females and males [45,46]. Sex-dependent differences in knee kinematics and neuromuscular control could also be observed [47]. It has been recommended to analyze sex-dimorphic landing mechanics data to understand differences in female and male ACL anatomy in the future in regard to estimating rupture risks [48]. In a recent study [49], the risk of rupture was 8.3 times higher in females than in males. Moreover, at the molecular level, the proteome of the ACL differed between female and male individuals: Comparing female with male ACL donors, alcohol dehydrogenase 1B and complement component 9 were more highly represented in female than in male. On the contrary, myocilin was the major protein detectable to higher degrees in males than in females [50]. The influence of estrogen on collagen in females might also be of interest in explaining sex-dependent higher risks of ACL injuries [51,52]. The menstrual cycle goes along with time-dependent differences in the biomechanical properties of the female ACL [53]. Not only this, in response to biomechanical challenges the ACL entheses comprise individual (influenced by personal loading history) architectures [1]. In addition to this, single-nucleotide polymorphisms of genes encoding for collagen correlated with the risk of non-contact ACL rupture in females [49]. The ACL can hypertrophy, showing increased cross-sectional areas in response to training in athletes [54]. However, the effect of extensive loading on the hypertrophy/remodeling of its enthesis is unclear. Individual aspects of the ACL were investigated to identify anatomical risk factors for rupture in males and females [55].

In regard to individual and sex-dependent risks, anatomical criteria such as the femoral notch diameter have been thoroughly investigated, with the hypothesis of a narrow notch, e.g., in females, being associated with higher risks of ACL failure, but not all published studies support this relation [22]. The posterior tibial slope was higher in females with ACL rupture compared to males [56].

### 2.6. ACL Animal Models

Animal models have been used extensively for experimental ACL reconstruction studies, including rabbit, pig, goat and sheep models. Since rodents (mice and rats) are too small, and in view of the costs of large animals, many of the ACL reconstruction studies are performed in rabbits [57,58]. Dogs, which are, in veterinary praxis, often patients with ACL rupture, are also sometimes used for experimental studies [59,60]. In view on these models, one should be aware of substantial species differences in the ACL anatomy, e.g., two anteriorly completely separately attaching anteromedial and posterolateral bundles in the porcine, caprine and sheep ACL models and larger dimensions of the pig ACL compared to that of humans [60,61,62]. Bascunan et al. concluded, based on their helpful review of comparative ACL anatomy and comparative biomechanical aspects, that the caprine and porcine models might show the closest similarity with human conditions [60]. However, the human knee is absolutely unique, e.g., in regard to tibial slope and plateau shape [60] and also concerning the ACL responding to loading during upright gait. Therefore, research approaches for anatomical ACL reconstruction using human ACL modeling/simulation data, e.g., derived from cadaver dissection, imaging or artificial human ACL models yet to be developed, would be very helpful in future.

### 2.7. In Silico ACL Modeling

In silico models based on human anatomical data sets from imaging, e.g., MRT or computer tomography, will become more important in future for research and clinical applications [63]. With an appropriate model of the knee for finite element (FE) analysis, it is feasible to simulate joint kinematics and kinetics. Furthermore, this enables statements to be made on failure behavior [64], reconstruction/fixation strategies [65,66] and implant design [67,68]. The development of adequate models is a complex and resource-intensive process. It requires a comprehensive workflow with data collection based on images and mechanical testing, an image segmentation step, surface geometry generation and meshing for finite element analysis as well as the definition of simulation settings [69]. Before the simulation step occurs, the model has to be customized by defining the material, loading and boundary conditions [70]. Since biological tissues exhibit highly anisotropic material behavior, the definition of material parameters and boundary conditions is a major challenge, especially when modeling the ACL enthesis as a connection between dissimilar tissue types (Table 1) over a small region. For comparison, bone is described as rigid and has a tensile strength of 120–180 MPa, a Young’s modulus of 15,000–25,000 MPa and a strain at failure between 2 and 3% [71]. In contrast, ligament tissue such as the ACL exhibits a tensile strength of 22–27 MPa, a Young’s modulus of 100–130 MPa and strain at failure between 25 and 30% [72].

There have been many mathematical and computational models, such as classical continuum structural mechanics approaches [73] or discrete fiber network models [74], developed focusing on different attachment modes with specific compositional and microstructural adaptations [75]. Hence, these modeling strategies address the mechanical role of fiber architecture [34,73,76], mineral reinforcements [77,78], surface roughness and interlocking [79].

Besides the different modeling approaches, the definition of the material properties is an essential point [77]. For this purpose, methods such as Fourier transform infrared spectroscopy imaging (FTIRI) are used to determine the collagen or proteoglycan content and the mineral distribution or extracellular matrix (ECM) composition [34,76]. Biomechanical investigations of the knee in terms of laxity tests in various degrees of freedom (DOF) [68] or traditional mechanical testing methods, such as uniaxial tensile tests, are required to determine characteristic properties, e.g., elastic modulus, tensile strength or displacement, and to ensure the correct prediction of knee motion and loading. Moreover, the anisotropic material properties of the ACL should be re-evaluated by full-field displacement measurements, which are independent of specimen geometry [69,80,81]. 

The body of literature focusing on the modeling and simulation of the ligament–bone complex with detailed information of the enthesis geometry and structural design is still lacking. Luetkemeyer et al. investigated the influence of ACL enthesis shape and attachment angle on the strain distribution in an FE model and demonstrated the possibility of making predictions in regard to injury risks [82]. Additionally, the constitutive model was further developed using full-field methods for ACL bundle modeling accounting for strain and principal material direction (collagen fiber splay) heterogeneity as well as enthesis shape [83]. Hence, it remains challenging to bridge the gap between clinical requirements and ACL modeling strategies, but this is essential for developing new substitutes for ACL TE approaches comprising the bone attachment. Machine learning could also be applied to predict cell behavior on scaffolds to reduce time-consuming and expensive experiments. This was demonstrated for cardiac tissue engineering [84]. Nevertheless, this approach needs sufficient data from the literature.

These models could be used to select optimized scaffold/cell compositions and develop a stable fixation variant for the tissue engineered graded scaffolds to rebuild the enthesis before starting with in vivo tests.

### 2.8. Strategies for Multiphasic Enthesis Scaffolds for ACL Tissue Engineering

To address the challenge of firm attachment of the ACL substitute onto the subligamentous bone, enthesis TE is required. To rebuild the graded structure of the ACL enthesis means creating a bi-, or better, triphasic topology of the cell carrier. Furthermore, a significant impact on enthesis formation can be achieved by using functionalized or bioactive materials. The latter should be a transient template with a degradation profile adapted to neotissue formation, but resilient enough as a tissue-engineered ACL substitute to withstand loading. So far, developed gradient scaffolds and helpful techniques for creating gradients in vitro have been reviewed previously [85,86]. To realize the concept of multiphasic scaffolds, several material and technical approaches were designed. Nevertheless, considering the radial hierarchical structure of the ACL (Figure 2), in addition to the longitudinal triphasic composition, a radial gradient could also be addressed in future as realized by the electrospinning of synthetic polymers for the reconstruction of other tissues [87].

#### 2.8.1. Material Composites

This section deals with different biomaterials for interface TE, mostly boiling down to a combination of materials for the bone and/or ligament zones, such as calcium phosphate ceramics (e.g., hydroxyapatite (HA) and β-tricalcium phosphate (β-TCP)) [88]; bioactive glasses (BG), e.g., silicate-based 45S5 Bioglass [89]; natural (e.g., silk [90,91,92,93,94], collagen and alginate) or even decellularized native matrices [95,96,97,98,99]; and synthetic (e.g., polycaprolactone (PCL), polylactic acid (PLA), polyglycolic acid (PGA) and polylactid-co-glycolide acid (PLGA) [100,101,102,103,104,105]) polymers. In addition, ways to combine different classes of materials, such as polymer–ceramic/BG composites [59,106], to adjust the mineral gradient or support bone integration will be highlighted. 

A decellularized enthesis ECM of the ACL would provide the best template for ACL TE [95,107] and also comprise the anatomical interdigitation of zones. Decellularization is needed to reduce immunogenicity which is associated with cells. Using a xenogenic decellularized tissue, interspecies differences in regard to zone dimensions, attachment angles, etc., have to be considered. The same is the case if tendon entheses are used as a matrix for ACL enthesis TE. The decellularized entheses of porcine calcaneal tendons were used previously [97] and implanted in rabbits, bridging medially (MCL removed) the knee joint space by passing through bone tunnels of the tibia and femur before being attached to connective tissue and tendon [97]. Decellularized tendon midsubstances were also prepared as ACL substitutes. To generate a graded template for ACL TE based on a tendon midsubstance, the decellularized ECM (rabbit flexor tendon) was treated with ultrasound to influence fiber arrangement leading to a random–aligned–random scaffold composite [98]. The latter facilitated the osteogenesis and chondrogenesis of MSCs and mediated superior integration in a rabbit ACL substitution model compared to the untreated control [98]. 

As another example, decellularized porcine superflexor tendons (to substitute the entire ACL) were used for ACL reconstruction in a sheep model. After implantation, the bone tunnel in the knees of sheep indeed showed Sharpey-like fibers attached to the bone and ossification of the graft within the bone tunnel as a features of bone integration was detectable [99]. In using decellularized templates, generally, toxic remnants and the wash out of essential sulfated glycosaminoglycans (sGAG) as well as growth factors (GF) have to be considered. In addition, the creation of pores that are accessible to cells and interconnective is needed without biomechanically weakening the ECM to allow sufficient reseeding of the scaffold zones with cells or even the immigration of cells in vivo, and this remains a mostly unsolved issue. An approach using laser perforation of decellularized cartilage to improve cell immigration into tracheal grafts was proposed by Baranovskii et al. [108]. It remains to be investigated whether such an engraving of a decellularized enthesis graft indeed stimulates ACL cell immigration without impairing its biomechanical properties. It could also lead to carbonized borders of the microchannels created by the laser, thereby affecting ECM adhesion motifs for ACL cells.

Collagen is the major ECM component of the ACL, and is also important in its enthesis and the major organic constituent of bone. Its fiber orientation changes in the attachment zones of the ACL (Figure 2). Representing a versatile tool for ACL scaffold functionalization [109,110], it can be chemically modified and crosslinked [111] to mimic the graded histoarchitecture of the ACL enthesis.

Bioactive materials that induce certain behaviors or processes such as osteogenesis are essential in the design of multiphasic structures. A hydrogel of silk fibroin was supplemented with ZnSr-doped β-TCP particles, showing some increase in the osteogenic activity of SaOs-2 cells with the aim to be added in future to the bone tunnels for ACL enthesis reconstruction [91]. The bioactive clay Laponite (LAP) and regenerated silk fibroin were used to fabricate hybrid fibers by wet spinning for woven ACL scaffolds. It was demonstrated that LAP improved osseointegration and the biomechanical properties by osteogenic induction [90]. Through the in vivo study by Han et al. using a biomimetic nanofiber membrane of PCL and HA, it was similarly proven that compared to unmodified PCL membranes, the bone integration and mechanical strength of the ACL graft improved, whereas fibrous scar tissue growth at the interface was inhibited [103]. In another experimental setting, the supplementation of implants with HA and collagen type I paste evoked some Sharpey fiber formation in canine ACL reconstruction as an indicator of successful ACL implant ingrowth [59]. Uehlin et al. investigated inkjet-printed HA nanoparticles to create a mineralization gradient for the ligament–bone interface. HA density influenced the cell morphology with more cuboidal-shaped cells in the high-density regions compared to more spindle-like shapes in the low-density regions to achieve a multiphase structure for enthesis TE [112].

BG was also used as an approach to facilitate osteogenesis in an ACL enthesis scaffold and the first very preliminary hints of an osteogenic cell response of MSCs were reported [106]. Furthermore, Spalazzi et al. demonstrated a triphasic scaffold system composed of PLGA and BG for interface TE of the ACL [105]. The triphasic scaffold supported cellular interactions, tissue infiltration and ECM production as well as mineralized and enthesis-like tissue regions in an in vivo study [105].

The advantages of natural polymers and bioactive materials are their high biocompatibility and osteoconductivity, while the disadvantages are the variability in properties due to their natural origin and limited availability. In contrast, synthetic polymers show adjustable biodegradability and beneficial mechanical properties but only limited bioactivity.

Therefore, combinations of different material classes should always be considered, as possible disadvantages of the individual components can be compensated for in ACL enthesis TE.

#### 2.8.2. Manufacturing Techniques

For positioning the cells in a zonal order, bioprinting might be the method of choice. Bioprinting is a technique that allows cells encapsulated in an ink to be placed at defined areas of a TE construct, with the ink mimicking the abundant ECM in the ACL and its enthesis. Combining functionalized inks allows adjustable gradients to be created as described by Neubauer et al. [113]. In this study, composites of recombinant spider silk hydrogels with fluorapatite rods were used to generate mineralized gradients. Subsequently, fibroblasts were encapsulated in the recombinant spider silk-fluorapatite hydrogels and gradually printed using unloaded spider silk hydrogels as the second component. Another approach was described for printing a promising ACL enthesis scaffold with metacrylated collagen and BG hydrogels forming a BG gradient. Raman spectroscopy was recommended as an effective method to validate the design and to compare it with the natural biocomposition of the ACL enthesis [106]. However, the limited biomechanical properties of so-far-available bioinks, mostly hydrogels, currently limits the capacity of this exciting technique in regard to ACL anthesis construct generation. A combination with 3D printing allows a defined scaffold topology to be created, mimicking the phases of the enthesis. Mimicking the histoarchitecture of the enthesis ECM requires, at the end, high-resolution manufacturing techniques comprising the nanostructure level. For the cartilage–bone interface, electrospinning with poly-lactic-co-glycolic acid (PLGA) and calcium nanoparticles was a helpful technique [114].

Accordingly, in another study, a triphasic bone–ligament scaffold consisting of PCL and PLGA was generated by electrospinning [115]. With collagen fibers being the main component of the native ACL and their typical an- and isotropic arrangement, nanofibers mimicking collagen might play a major role in cell instruction. Electrospinning allows for the deposition and coordination of nanofibers [116,117]. It also allows different synthetical polymers to be combined in a multilayer, and the biomechanical properties of the resulting multilayer are influenced by the degradation profiles of the components and their position (outer or inner layer) [87].

Xiong et al. created a bone–ligament interface by melt electrospinning/writing nicely simulating the crimped fiber micropattern for the ligament area to instruct fibroblasts and crimped small fiber spacing, inducing calcium deposition by osteoblasts [118]. Referring to the natural ACL ECM, the crimp pattern is more complex with highly and lesser-crimped fiber bundles [22], which is to be addressed in future.

As another option for ligament/bone interface TE, a biphasic scaffold was fabricated by a freeze-drying strategy using silk fibroin. In this approach, the collagen fiber orientation of the natural enthesis was mimicked [41].

Braiding and knitting are textile techniques to create biomechanically resilient scaffolds. Sophisticated triphasic scaffolds based on a knitted silk fibroin structure were prepared for ACL reconstruction [13,14] and either seeded with bone-marrow-derived MSC (BMSCs), chondrocytes, osteoblasts (this silk scaffold variant was zone-dependently modified with hyaluronic acid or chondroitin sulfate) [13] or, in another approach of the research group, lentivirus vectors encoding either TGFβ3 and BMP-2 (including an HA-coated scaffold area) were immobilized on separate zones of a similar silk scaffold before being seeded with BMSCs to induce their zone-specific differentiation [14]. Implantation in a rabbit ACL substitution model showed integration into bone with a multilayered tissue structure [13]. Kimura et al. braided a hybrid PLA scaffold with different wrapped membranes composed of gelatin hydrogel and collagen (plus growth factor bFGF) to generate a triphasic structure [119]. The mechanical properties were enhanced, and collagen production, osseointegration, intrascaffold cell migration and vascularization of regenerated ACL tissue were improved [119]. The additional integration of non-degradable polyethylene terephthalate (PET) fibers in braided scaffolds investigated by Mengsteab et al. showed an enhanced osteointegration, illustrating the potential of braiding as a manufacturing process [120].

The embroidery technology was applied as a tool to produce a triphasic ACL enthesis scaffold using a combination of poly(L-lactide-co-ε-caprolactone) (P(LA-CL))/PLA. By the embroidery pattern defined, porosity, pore size and pore and fiber alignment could be established to create a graded construct, and biomechanical requirements can be simultaneously addressed by the design. The interdigitation between zones can also be integrated by the embroidering technique [109]. In addition, it has been shown that embroidered scaffolds could be modified by targeting a collagen substitute to act as a temporary cell barrier and thus potentially as a tidemark [121]. The tidemark as a cell barrier was also addressed and integrated as a collagen membrane into embroidered scaffolds [121]. However, its anatomy as a 3D undulating few-microns-thick laminar structure [18] could not be recapitulated by this approach. A comparison of the previously mentioned textile technologies reveals the following drawbacks: braided structures tend to have low porosity, knitted fabrics have comparatively low stability and the embroidery process has a limited ability to produce tubular structures. In addition, the use of fibers with sufficient stability is always a prerequisite for the manufacturing process. Nevertheless, these technologies allow the material- and time-efficient fabrication and development of biomechanically stable and multiphase scaffolds for the TE of the ACL.

It has to be discussed in future that the natural transition of the phases of the enthesis is not plane and abrupt, but interdigitations [109] can be found, increasing its natural stability. In most of the triphasic scaffolds, this aspect has not been realized yet. Moreover, in response to the natural ECM remodeling process in tissues induced by changed loading, the pattern of transition zones in the ACL enthesis might also change. Scaffold topology created by the above-mentioned techniques is doubtless important to instruct cells to assume a zone-specific synthetic active phenotype, but might not be sufficient. 

### 2.9. Biochemical Instruction of Cells Used for ACL Tissue Engineering

Different factors have been applied to enforce the enthesis lineage differentiation of different cell types (Table 2). Factors were either added separately or integrated into the scaffold. As an early in vitro approach to address enthesis formation, a biphasic scaffold was generated and seeded with fibroblasts consisting of a fibrin gel with TGFβ (subtype not specified) and brushite supplementation to form the enthesis part. An enthesis-like tissue formation with a border between both parts resembling a tidemark could be detected [122]. A more recent study also based on a hydrogel described a tendency of multilineage differentiation of MSCs through the spatial presentation of growth factors (BMP-4 and TGFβ3) in a multiphasic gelatin-methacrylate hydrogel for enthesis cell type differentiation [123].

For tendon/ligament reconstruction TGFβ2 and GDF5 were zone-specifically tethered to a biphasic silk fibroin scaffold. In addition, anisotropic (ligament part) and isotropic (bone/cartilage part) pore alignment was arranged. TGFβ2 alone triggered in adipose-tissue-derived stem cells (ASC) the expression of ligament-related factors, whereas in combination with GDF5 and an isotropic scaffold topology, it favored cartilage marker expression [124]. In another study, lentiviral vectors encoding TGFβ3 and BMP-2 were selected and integrated using phosphatidylserine as an anchor for silk scaffold fibrocartilage and bone zones in ACL constructs to support enthesis cell differentiation [14]. TGFβ2 and TGFβ3 have time-dependent effects [125]. Some lineage-related differentiation of MSCs in response to TGFβ3 and BMP-2 could be shown by Fan et al. [14]. A poly-L-lactic acid scaffold with a bone and ligament compartment was braided and supplemented with BMP-2 before being tested in the rabbit ACL reconstruction model. The BMP-2 supplementation led to a trend of smaller bone tunnel cross-sectional areas [57]. 

Acidic fibroblast growth factor (FGF)2, combined with collagen, was added to the bone tunnel in which autografts in a rabbit ACL reconstruction model were fixed to support bone integration. This treatment induced well-organized Sharpey fiber and a fibrocartilage transition zone formation as a sign of firm integration [58].

In a comparable approach (rabbit ACL reconstruction model), osteoprotegerin (OPG) combined with deproteinized xenograft bone were installed into the bone tunnel. It indeed improved the bone integration of the tendon grafts compared to the groups with no OPG/bone treatment [126].

Despite another study not representing a true ACL implantation approach, it was interesting to see that hepatocyte growth factor (HGF) led to enthesis-like tissue formation in a tendon bone healing model in rabbits [127]. Looking at the promising results of the above-listed studies, one has to bear in mind that the rabbit model is well known to exert a high healing potential, which is possibly not fully transferable to human conditions.

**Table 2 ijms-24-09745-t002:** Factors used for cell instruction to mimic ACL enthesis zones.

Factor	Aim/Purpose	Result	Reference
BMP-4 and TGFβ3	Spatial presentation of growth factors (captured in a methacrylate hydrogel construct) with single-concentration and gradient areas for inducing an enthesis phenotype in MSCs	Resulting in a spatially heterogeneous response	[123]
TGFβ3 and BMP-2	Letiviral vectors integrated should stimulate fibrocartilage and bone zone differentiation of MSCs	Lineage differentiation could be detected, firm integration in vivo.	[14]
aFGF2	Improve integration of an ACL substitute into the bone tunnel	Improved Sharpey fiber formation	[58]
Heparin	Regulation of the release of BMP-2 and TGFβ3	Heparin kept growth factors locally biologically active at low doses	[124]
SDF1	Improve bone integration of collagen silk scaffolds by SDF-1 release	Controlled SDF1 release (7 d) promoted bone tunnel integration of silk–collagen scaffolds in vivo, attraction of ligament precursor cells (separately injected in the rabbit joint)	[128]
TGF-β2 and GDF5	Generate a ligament construct with enthesis-like tissue	TGFβ2 and pore anisotropy synergistically elevated ligament markers and collagen type I/enhanced the expression of cartilage markers and collagen II protein content on substrates with isotropic porosity	[124]
Ascorbic acid, proline, TGFβ (subtype not specified)	Collagen induction in tendon/ligament part	Increased collagen content, with further increase from application of TGFβ	[122]
Brushite	Generate a hard tissue zone in an in vitro ligament model	Interface strength increased, tidemark-like transition between tendon/ligament and enthesis parts	[122]
Osteoprotegerin (OPG)	To improve bone integration of a tendon graft for ACL substitution using OPG combined with deprotinized bone	Higher amount of Sharpey fiber formation	[126]

Cellular and biochemical gradients can be applied using a novel bioreactor [129]. There are other promising factors which have so far not been investigated in much detail for ACL enthesis TE: Periostin, which seems to contribute to fibrocartilage layer growth, and is involved in ACL enthesis fibrocartilage layer formation [130,131], could be of major importance; moreover, distinct splice variants of periostin seem to be indicative either for ACL precursor cells or chondrocytes [132]. Additionally, cell stimulation, which has not been used for enthesis TE but for ACL, with mechanogrowth factor (MGF), which is an IGF1 splicing variant, is another candidate [133]. It stimulated ACL fibroblast proliferation and collagen synthesis [134,135].

### 2.10. Ligamentogenic/Entheseal Differentiation of Cells for ACL Enthesis Tissue Engineering

Ligament fibroblasts (ligamentocytes) have a slow growth rate [136] and are difficult to access. Theoretically, they could be harvested from the stumps of a ruptured ACL, isolated and after sufficient cell expansion be used for the TE construct. A more accessible cell source with full ligamentogenic differentiation potential might be iliotibial-band-derived fibroblasts [137].

The plasticity of tendon cells is known; when the tendon goes around a pulley, fibrocartilage develops in response to compressive loading [138]. In tendons, different stem cell niches have been shown [139] which could also be hypothesized for ligaments. It remains a point of discussion whether resident stem cell populations are responsible for aberrant lineage differentiation mediating tissue metaplasia in pathological ACLs [25,140], and whether enthesis neoformation in implanted grafts or even differentiated ligament fibroblasts can transdifferentiate. By single-cell sequencing, Fang et al. showed a stem cell pool (Gli+ progenitor population) important for tendon enthesis formation and healing [141]. This Gli1+ progenitor cell population can differentiate into fibrocartilage-producing cells [142]. This cell population remains to be shown for the ACL.

ASCs were easily isolated from lipoaspirates [136,143], and hence represent an attractive source for musculoskeletal TE. However, the in vitro supplementation of human ASCs with anabolic growth factors (TGFβ1, insulin-like growth factor (IGF), epidermal growth factor (EGF) and FGF were tested) was not sufficient for commitment into the ligamentogenic lineage [136]. On the contrary, ASC cell sheets promoted ligament bone healing in a rabbit model [144]. ASCs applied to tendon enthesis scaffolds supplemented with HA on the bone and chondroitinsulphate on the ligament part expressed the typical ECM components of tendon in the tendon zone with the aligned fibers, or bone tissue in the random mineralized part [38]. In another study, ASCs changed their expression profile depending on the pore alignment in multiphasic enthesis scaffolds [41]. The autologous stromal vascular fraction of ASCs, an adipose-derived, stem cell-rich isolate from knee fat pads, was also used to seed silk fibroin scaffolds which were subsequently successfully tested in an ovine ACL reconstruction model. However, the implantation of the cell-seeded scaffolds did not lead to histologically different results compared to cell-free implantation in regard to bone integration [92,93].

Many of the aforementioned ACL TE studies based on BMSCs were tested on a multiphasic scaffold dotted with GFs which promoted their lineage differentiation [14].

Interesting is the observation that even skin fibroblasts seeded on collagen scaffolds and implanted in a goat model led to formation of enthesis-like tissue in vivo with Sharpey fibers and fibrocartilage [145]. Nevertheless, it remains unclear whether fibroblast plasticity or the trophic factors released by them triggered the observed tissue formation.

### 2.11. Co-Cultures for Zonal ACL Tissue Engineering and Mechanical Training of Cells

Several co-culture approaches have been reported for ACL tissue-engineered constructs. A trilineage co-culture system (osteoblasts–BMSCs–ligament-derived fibroblasts) was established on a hybrid silk scaffold. Particularly, the influence of TGFβ3 was tested, but seemed rather to inhibit the mineralization [146]. A dual-chamber system was used with blended fibrin–alginate hydrogel scaffolds for ligament-interface TE. The endochondral, fibrocartilaginous or ligamentous attachments were triggered by the spatial exposition of the same cell type (MSCs) to growth factors such as TGFβ3 [94]. Mesenchymal stem cells (BMSCs), chondrocytes and osteoblasts were co-cultured on triphasic silk enthesis scaffolds. The construct showed a successful integration into bone in the rabbit ACL reconstruction model [13]. Fibroblasts and osteoblasts were co-cultured on a triphasic scaffold (melt-electrospun/written) with different grid crimp microtopologies, which supported lineage differentiation [118].

In the cases of co-culturing zone-related cell types of the enthesis on the same area, directed seeding strategies are required. Different approaches to achieve an exact position of the cells on the constructs have been applied. The usage of spheroids is a method to bring cells into porous scaffolds. Cells stored in the spheroids emigrate on the carrier. The size of the spheroids could be adapted to the pore size. It is known that ACL ligamentocytes maintain their expression profile in spheroids [147,148,149,150,151]. Using cell sheets is another tool to place cells on a defined position on the scaffolds, leading to cell emigration on them [152]. Capturing cells within a hydrogel and fixing them in the scaffold presents another tool of directed seeding. This could be combined with bioprinting, which allows cells or cell aggregates to be placed exactly in the area wanted. Cells can be mixed into a hydrogel; however, the contact to the scaffold material is not given in the beginning, and it could happen that cells fall out and are lost when the hydrogel starts degradation. Another challenge in co-culturing cells with different proliferative activity might be the overgrowth of zones by one cell type. Hence, the concept of a transient barrier integrated into scaffolds to inhibit cell overgrowth was proposed [121]. This barrier should allow nutrition to pass through and disappear at the time point that it is not required anymore.

Although a more detailed insight would exceed the scope of this review, the importance of mechanical training needs a short remark. The detrimental effect of unloading on the tendon enthesis has been thoroughly discussed in the review of Roffino et al. [153]. In contrast to tendon/tenocytes, for ligaments/ligamentocytes fewer studies are provided [15,154], and particularly for ACL cells only a few published studies discussing and presenting suitable protocols for cell training are available [150,155]. Nevertheless, since ligaments and tendons share many of their anatomical and physiological properties, results from tendon can also be used as a template for ligaments. However, the complex biomechanical environment of the ACL enthesis is difficult to mimic. It comprises a cyclic dynamic loading environment of combined tensile, compressive and shear forces [9]. To account for stimulatory factors released by cells in response to mechanostimulation, an ACL TE model using exercise-conditioned human serum was proposed [156].

## 3. Materials and Methods

A medline study was undertaken with search mesh terms such as ACL combined with enthesis, ligament, fibrocartilage and reconstruction. Hereby, the focus was on the human ACL. As far as possible, the selected literature was restricted to the ACL and did not comprise rotator cuff and entheses of other ligaments or tendons.

## 4. Conclusions and Perspectives

The ACL entheses comprise individual (loading history) asymmetric and polar histoarchitectures. They result from the unique biomechanical microenvironment of overlapping tensile, compressive and shear forces involved in enthesis formation, maturation and maintenance. Since the ACL enthesis comprises tissues rich in ECM and the synthesis of a biomechanically competent ECM needs a lot of time, transient cell carrier structures (“scaffolds”) are needed as an essential initial template for TE of the ACL enthesis. To guide tissue formation, the scaffold must be biomimetic and multiphasic. In addition, the biomechanical properties must be appropriately designed to be comparable to those of the native enthesis tissue, and the spatial topology, such as zone-dependent porosity, pore sizes and fiber orientation, must be taken into account in the scaffold design, as these can have a direct influence on cell behavior.

It is difficult to guide the regeneration process of the enthesis due to its complex tissue structure, with fibro-, chondro- and osteogenesis needed to be addressed spatiotemporally. Key parameters of the current approaches include the material selection and scaffold fabrication technology, the targeted selection of specific cell types and appropriate growth factors and biophysical modulation. Each of these issues is considered as a separate field of research and requires extensive investigations, both to specify the parameters themselves and to coordinate the interplay of diverse influencing factors in tissue formation. 

This review has highlighted the main material compositions and fabrication techniques for ACL enthesis TE. As a result, there is an urgent need for improved biomimetic hierarchical structures that meet the aforementioned requirements. To this end, future research should be conducted on technology combinations, e.g., textile macroscale substitutes as a mechanically stable unit together with electrospinning/writing to generate micro to nano substructures. The enthesis fibrocartilage phases of the human ACL have microdimensions of only ~0.4–1 mm. Designing an enthesis gradient scaffold for testing in an animal model, e.g., rabbit, might be even more challenging. The integration of the natural interdigitation of gradient zones should be an issue in future scaffold design. Moreover, considering the sub-bundle anatomy in scaffold design, recapitulating the orientation of anteromedial and posteromedial strands of the human ACL should be taken into account.

Furthermore, the combination of cells capable of enthesis-specific lineage differentiation with supportive growth factors or other biostimulation strategies to establish a native ACL enthesis are required. Members of the TGF superfamily growth factors, such as TGFβ, BMP and GDF subtypes and isoforms, have been tested separately or combined for ACL enthesis TE with some lineage-related instructive effects; however, groundbreaking efforts are still lacking due to limited knowledge concerning the best GF candidates, their optimal release profile and effective concentrations. In addition, the selection of a cell source, fully capable of generating enthesis zone-specific cell populations in vivo, e.g., ASCs or MSCs, which are intensively researched and are easy to harvest and propagate, remains a challenge. In the case of combining different cell types or precursor cells committed to ACL zone-specific differentiation lineages, directed seeding strategies are needed, e.g., based on cell sheets or spheroids [151,152], and a mechanostimulation profile closely mimicking the unique biomechanical niche of the ACL enthesis. The development of modeling approaches for the human ACL enthesis might be very helpful in future to address some of the above-mentioned open questions, particularly concerning scaffold biomechanics.

However, the big challenge is to transfer excellent TE research into application. With its increasing complexity, its approval as a medical device becomes more difficult and thus also increases the costs, which are of immense importance for medical device manufacturers, but also clinical users and health insurance associations, when investing in new procedures and products.

Taken together, a couple of promising approaches of ACL enthesis TE are available, comprising graded zonal scaffolds combined with co-cultures of different precursor cells; however, rebuilding the complex anatomy and histoarchitecture of the ACL remains a challenge.

## Figures and Tables

**Figure 1 ijms-24-09745-f001:**
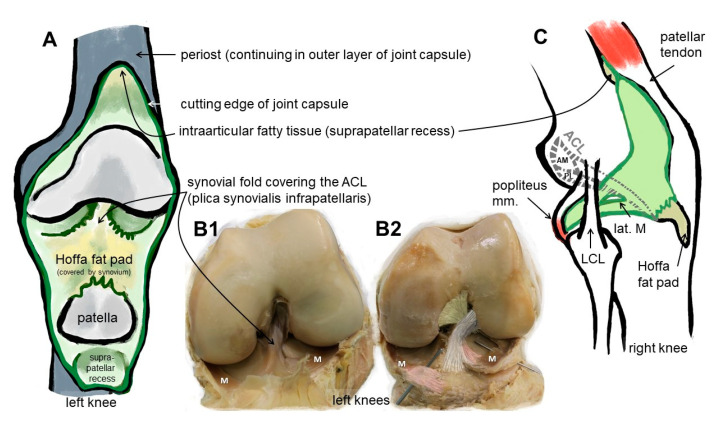
Macroscopical anatomy of the human ACL. Scheme of the insight into the knee joint cavity after the joint capsule of the knee is opened and its inner surface covered by the synovial membrane (except for the articulating surfaces) is shown (**A**). (**B1**) shows the same gross anatomical image and (**B2**) visualizes by coloration the relation of the ACL with anteromedial (beige) and posterolateral bundle (grey), menisci (M) and anterior meniscotibial ligaments (rose). (**C**) The extension of the synovial membrane is shown (lateral view, fibrous capsule layer removed). ACL: anterior cruciate ligament, AM: anteromedial bundle, lat.: lateral, LCL: lateral collateral ligament, M: meniscus, m: muscle, PL: posterolateral bundle. The image was created by G. Schulze-Tanzil using krita 4.1.7.

**Figure 2 ijms-24-09745-f002:**
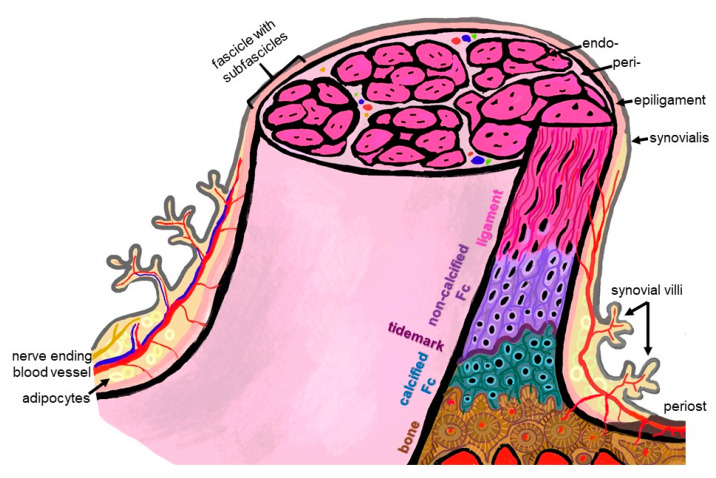
Scheme summarizing some histological features of the ACL enthesis. The ACL, covered by the synovial layer and embraced by the epiligament, is hierarchically organized in bundles (surrounded by the periligament) and sub bundles (encircled by the endoligament). Epi-, peri- and endoligament sheets represent flexible connective tissue layers, which also contain blood and lymphatic vessels as well as nerves. The enthesis comprises a graduated attachment into bone: starting with the ligament phase with increasing glycosaminoglycan/proteoglycan contents followed by non-vascularized and non-calcified fibrocartilage (FC), a tidemark forming the border to the calcification front of the calcified FC also lacking blood vessels down to the vascularized bone. The phases thoroughly interdigitate with each other. From ligament to bone, the ECM composition and collagen fiber quality and alignment change from anisotrop to isotrop. The image was created by G. Schulze-Tanzil using krita 4.1.7.

**Table 1 ijms-24-09745-t001:** Zonal expression profile of fibrocartilaginous enthesis (color scheme of the zones according to Figure 2).

Zone	Cells and Morphology	Components	Property	Stiffness	Ref.
**Ligament**	Elongated ligamentocytes, partly ovoid, in rows, low cellularity, capillaries	COL-I, aligned, decorin, biglycan	Anisotrop	242 ± 28 N/mm 65.3–111 MPa *	[36,38,39,40]
**Fibrocartilage non-calcified**	Ovoid fibrochondrocytes, groups, in rows, basophilic hollow, higher cellularity than midsubstance, no capillaries	COL-II, -III, -IX, aggrecan, (decorin)	Transition	<0.50 MPa	[36,38,41,42]
**Tidemark**			Barrier-like	data not available	[19,20,21]
	(No cells) ~5 µm, trilaminar basophilic line, marks calcification front	Phospholipids, alkaline phosphatase, adenosine triphosphatase, lectins, high mobility group box chromosomal protein 1 (HMGB1), lead, zinc and tetracycline	Intense ECM staining	data not available	[38,43,44]
**Fibrocartilage** **calcified**	Round fibrochondrocytes, lacunae, partly hypertrophic, higher cellularity than midsubstance, no capillaries	COL-I, -X, -II, HA (gradient) aggrecan, alkaline phosphatase	Transition	≥0.5 MPa *	[34,35,36,38,41,42]
**Bone transition**	Osteoblasts and -clasts, osteocytes, capillaries	COL-I, HA (high)	Isotrop	18,203 MPa	[38,40]

COL, collagen, HA: hydroxyapatite, * M. supraspinatus tendon.

## Data Availability

Supporting data can be obtained from the authors on request.

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
