# Peer review of "Anatomical Tissue Engineering of the Anterior Cruciate Ligament Entheses"

_ijms, 2023, doi:10.3390/ijms24119745_

Round 1

Reviewer 1 Report

Review

Title: Anatomical tissue engineering of the anterior cruciate ligament entheses

General

The authors give a good overview on the anatomical challenges of tissue engineering a ligament- bone enthesis. After some minor revisions, I suggest this review article for publication.

Specific

Line 26 instead of [BMP]2, better BMP-2

Table 1 Could the zones be depicted adjacent to the corresponding parts of the enthesis, so that the tidemark is actually side-by-side with the word tidemark?

Moreover, it would be good to have a further column in table 1 with the approximate values for the stiffness of each part of the enthesis. In the text, it is mentioned that the stiffness increases from the ligament to the bony part.

Line 222 it should be neuromuscular (typo)

Line 234 Please comment on the risk factors for rupture that are different in male and female.

Line 249 after Bascunan et al. no comma

Line 254 there is twice “future” in the same sentence; please omit it once.

Lines 356 ff. electrospinning of gradient tissue engineered constructs; please cite the following: Cartilage/bone interface fabricated under perfusion: Spatially organized commitment of adipose-derived stem cells without medium supplementation by W. Baumgartner, L. Otto, S. C. Hess, W. J. Stark, S. Märsmann, G. M. Bürgisser, et al. in Journal of Biomedical Materials Research Part B: Applied Biomaterials 2018 Vol. 107B Pages 1833–1843.

Line 375 Elaborate on the barrier function of the tidemark.

Line 387 How could interdigitation be realized best? With what technique of the ones presented?

Table 2 summarizes studies with TGF-b2 and TGF-b3. Comment on the differential effects of these two growth factors in the main text.

Line 446 mechanogrowth factor MGF, please mention that this is an IGF-1 growth factor and cite In   Vitro and In Vivo Effects of IGF‑1 Delivery Strategies on Tendon Healing: A Review  by I. Miescher, J. Rieber, M. Calcagni and J. Buschmann in International Journal of Molecular Sciences 2023 Vol. 24 Issue 3 Pages 37.

Line 465 ASCs are easily isolated from lipoaspirates (instead of were). Please cite: Yield and proliferation rate of adipose-derived stem cells as a function of age, BMI and harvest site: Increasing the yield by using adherent and supernatant fractions? By J. Buschmann, S. Gao, L. Härter, S. Hemmi, M. Welti, C. M. L. Werner, et al. in Cytotherapy 2013 Vol. 15 Issue 1 Pages 1098-1105.

Line 480 after BMSCs no point

Some few typos.

Author Response

Dear Ladies and Gentlemen,                                                                Nuremberg, 12th May 2023

Dear Editor,

The authors would like to thank the reviewers for carefully reading the manuscript and their very valuable comments. We modified the manuscript according to the reviewers suggestions with a list of changes shown below. All corrections and addenda performed are indicated in red in the revised version of the manuscript. We hope you will find this manuscript suitable for publication in the “International Journal of Molecular Science”. Please do not hesitate to contact me anytime for questions regarding this manuscript. Several orthographic mistakes were corrected and figures were optimized.

Sincerely,

Univ.-Prof. Dr. Gundula Schulze-Tanzil

Authors‘ point by point responses to the reviewer‘s comments:

Reviewer 1:

General

The authors give a good overview on the anatomical challenges of tissue engineering a ligament- bone enthesis. After some minor revisions, I suggest this review article for publication.

Response: The authors would like to express their sincere thanks to the reviewer for taking the time to accurately evaluate this manuscript and provide us with constructive feedback. The authors have tried to address (red marked and highlighted in the text) all ambiguities and suggestions for improvement of the manuscript.

Specific

Line 26 instead of [BMP]2, better BMP-2

Response: We added the hyphen, but due the fact, that this is the first mention of the word we would not immediately use the abbreviation. Nevertheless, we used the hyphen for the BMP subtypes now throughout the manuscript.

Table 1 Could the zones be depicted adjacent to the corresponding parts of the enthesis, so that the tidemark is actually side-by-side with the word tidemark?

Response: We have tried to optimize the Table 1 and as you see now the “Tidemark” is better positioned to the corresponding text, but it will not work with the structures beneath, like the calcified cartilage or the bone zones. That´s why we have created an additional picture with the description. We hope that the new figure (Figure 2) will satisfy your requirements and is now easier to understand.

Moreover, it would be good to have a further column in table 1 with the approximate values for the stiffness of each part of the enthesis. In the text, it is mentioned that the stiffness increases from the ligament to the bony part.

Response: This column was added to table 1 and filled with so far available information. However, data for the ACL zones are incomplete and hence, we had to add information from other entheses marked with an asterisk and footnote in the table.

Line 222 it should be neuromuscular (typo)

Response: We thank the reviewer and corrected the mistake.

Line 234 Please comment on the risk factors for rupture that are different in male and female.

Response: We added references supporting the higher risk in females and underlined discussed anatomical criteria.

Line 249 after Bascunan et al. no comma

Response: The comma has been deleted.

Line 254 there is twice “future” in the same sentence; please omit it once.

Response: We corrected the sentence.

Lines 356 ff. electrospinning of gradient tissue engineered constructs; please cite the following: Cartilage/bone interface fabricated under perfusion: Spatially organized commitment of adipose-derived stem cells without medium supplementation by W. Baumgartner, L. Otto, S. C. Hess, W. J. Stark, S. Märsmann, G. M. Bürgisser, et al. in Journal of Biomedical Materials Research Part B: Applied Biomaterials 2018 Vol. 107B Pages 1833–1843.

Response: Thank you for the recommendation – we cited this manuscript now (section 2.8.2, line 401).

Line 375 Elaborate on the barrier function of the tidemark.

Response: Overall, the function of the tidemark remains unclear. We mention that the tidemark is a barrier between calcified and non calcified fibrocartilage in section 2., and explain its relation to the nutrition fronts per diffusion an added it in table 1 (property: barrier-like). It was also mentioned in the scaffold part.

Line 387 How could interdigitation be realized best? With what technique of the ones presented?

Response: Embroidering is a suitable approach. It was realized (Gögele et al., 2023). Decellularized enthesis scaffolds could also be used as reviewed by Lei et al., (Lei et al., 2021). It was added now in 2.8.2.

Table 2 summarizes studies with TGF-b2 and TGF-b3. Comment on the differential effects of these two growth factors in the main text.

Response: We cite Ferguson and O´Kane (2004) now to underline the spatial- and time-dependent effects of the TGF isotypes.

Line 446 mechanogrowth factor MGF, please mention that this is an IGF-1 growth factor and cite In Vitro and In Vivo Effects of IGF‑1 Delivery Strategies on Tendon Healing: A Review by I. Miescher, J. Rieber, M. Calcagni and J. Buschmann in International Journal of Molecular Sciences 2023 Vol. 24 Issue 3 Pages 37.

Response: The above mentioned publication was cited. We explain that mechanogrowth factor is an IGF1 splicing variant.

Line 465 ASCs are easily isolated from lipoaspirates (instead of were). Please cite: Yield and proliferation rate of adipose-derived stem cells as a function of age, BMI and harvest site: Increasing the yield by using adherent and supernatant fractions? By J. Buschmann, S. Gao, L. Härter, S. Hemmi, M. Welti, C. M. L. Werner, et al. in Cytotherapy 2013 Vol. 15 Issue 1 Pages 1098-1105.

Response: The above mentioned publication is helpful and was cited (2.10, lines 517-519).

Line 480 after BMSCs no point

Response: The point was deleted.

Reviewer 2 Report

The presented review provides a road map of key parameters to be considered in future in anterior cruciate ligament (ACL) interface tissue engineering (TE) approaches.

The paper addresses a specific gap in the field by providing a comprehensive review of the pecularities of synovioentheseal complexes at the femoral and tibial attachment sites, as well as discussing emerging tissue engineering strategies that address them. However, the approach used by the authors to describe the roadmap of key parameters is completely insufficient. The findings in strategies to manufacture multiphasic enthesis scaffolds appear to be trivial.  

The paper provides the approach of anatomical tissue engineering that is fairly new and little-studied in the field.   The paper schuld be improved by including the responses for the issues: 1. The effects of different loading approaches on enthesis formation, maturation and maintenance; 2. The impact of various material composites and manufacturing techniques on zonal cell carrier formation. 3. The meaningful influence of functionalized or bioactive materials, growth factors, and biomechanical microenvironments on enthesis properties. 4. Combinations of external parameters caused affects of ACL graft integration. The references are out of date. I recommend authors use more sources 2017-2022. The papers look like a part of the scientific report, not a review paper. I recommend authors completely revise the paper.  

Author Response

Dear Ladies and Gentlemen,                                                                Nuremberg, 12th May 2023

Dear Editor,

The authors would like to thank the reviewers for carefully reading the manuscript and their very valuable comments. We modified the manuscript according to the reviewers suggestions with a list of changes shown below. All corrections and addenda performed are indicated in red in the revised version of the manuscript. We hope you will find this manuscript suitable for publication in the “International Journal of Molecular Science”. Please do not hesitate to contact me anytime for questions regarding this manuscript. Several orthographic mistakes were corrected and figures were optimized.

Sincerely,

Univ.-Prof. Dr. Gundula Schulze-Tanzil

Authors‘ point by point responses to the reviewer‘s comments:

Reviewer 2

The presented review provides a road map of key parameters to be considered in future in anterior cruciate ligament (ACL) interface tissue engineering (TE) approaches.

The paper addresses a specific gap in the field by providing a comprehensive review of the pecularities of synovioentheseal complexes at the femoral and tibial attachment sites, as well as discussing emerging tissue engineering strategies that address them.

However, the approach used by the authors to describe the roadmap of key parameters is completely insufficient. The findings in strategies to manufacture multiphasic enthesis scaffolds appear to be trivial.

Response:  The previous focus of the manuscript was on the peculiarities of the ACL entheses and not on materials and techniques for scaffolds for enthesis TE. According to the reviewers criticism we broadened the focus including more detailed. Section 2.8.1 (material composites) was added, braiding, knitting, embroidering was critically discussed in 2.8.2 (manufacturing techniques). Mechnical training was added (lines 591-602). The conclusion section (4., lines 608-660) was thoroughly elaborated summarizing the unmet medical needs.

The paper provides the approach of anatomical tissue engineering that is fairly new and little-studied in the field. The paper schuld be improved by including the responses for the issues: 1. The effects of different loading approaches on enthesis formation, maturation and maintenance; 2. The impact of various material composites and manufacturing techniques on zonal cell carrier formation. 3. The meaningful influence of functionalized or bioactive materials, growth factors, and biomechanical microenvironments on enthesis properties. 4. Combinations of external parameters caused affects of ACL graft integration.

Response:  To fulfil the disappointed expectations of the readers we added now a deeper discussion of biomechanical aspects (table 1, column „stiffness“), materials (2.8.1) / techniques (2.8.2) for zonal scaffold approaches, functionalization/bioactivity cues and integration for ACL TE.

The references are out of date. I recommend authors use more sources 2017-2022.

Response:  The pathbreaking anatomical facts concerning the ACL enthesis have been investigated a couple of years ago and hence, have to be cited despite being out of date. Therefore, the impression arises that some references are out of date. Nevertheless, many novel articles have been integrated now (e.g. Miescher et al., 2023, Luo et al., 2022; Ribeiro et al., 2022, Uehlin et al., 2022).

The papers look like a part of the scientific report, not a review paper. I recommend authors completely revise the paper. 

Response: In the course of the revision the anatomical microarchitecture of the ACL enthesis were discussed in greater detail (novel figure 2) including sex-dependent and individual pecularities as well as additional details concerning biomechanics and approaches to establish zonal and biofunctional ACL scaffolds were added now. The authors have tried to further elaborate and improve all sections of the manuscript.

“The paper provides the approach of anatomical tissue engineering that is fairly new and little-studied in the field. The paper schuld be improved by including the responses for the issues:

  1. The effects of different loading approaches on enthesis formation, maturation and maintenance;”

Response: We added a sequence about the mechanical stimuli and their influence on the enthesis properties during formation, maturation and maintenance (2.11). It is a bright field and hence, could be a novel topic of a novel review. Therefore, it can not be elaborated in full detail.

“2. The impact of various material composites and manufacturing techniques on zonal cell carrier formation.”

Response: We comprehensively reviewed chapter 2.8. with new subchapter about material composites and techniques.

“3. The meaningful influence of functionalized or bioactive materials, growth factors, and biomechanical microenvironments on enthesis properties.”

Response: It was included in new subchapter 2.8.1

“4. Combinations of external parameters caused affects of ACL graft integration.”

Response: We revised the introduction section adding considerations about ACL graft integration (lines 81-90).

“The references are out of date. I recommend authors use more sources 2017-2022.”

Response: We have placed great emphasis on the use of current literature and about two-thirds of the literature derives from the time period mentioned. In addition, we have now specifically researched and cited literature from 2021-2023 for the revision and integrated into the additional novel sections of the manuscript. However, it is important to note that the fundamental findings on the anatomy of the ACL enthesis are based on studies before 2017, so citation of these pathbreaking findings cannot be omitted.

“The papers look like a part of the scientific report, not a review paper.”

Response: We have revised the manuscript according to the recommendations of reviewer 1 and 2 extensively and addressed this point by expanding the conclusion section to include a classification of the review and highlighting perspectives for the research field.

“I recommend authors completely revise the paper.”

Response: We completely revised the paper and added new essential information according to the reviewer suggestions.

Round 2

Reviewer 2 Report

The authors responded satisfactorily to all my comments and made the necessary changes to the manuscript. However, after reading the revised manuscript, a few questions came to my mind. 1. Mechanical properties of scaffolds define the physiological relevance of formed implant. Some bioengineering strategies can be applied to achieve the physiological relevance of the implant. I recommend discussing some novel ideas in this area [https://doi.org/10.3390/biomedicines11030745] [https://doi.org/10.1016/j.compbiomed.2023.106804]. 2. Another part that should be discussed in the paper is cellular compatibility, i.e. the ability of cells to seed and mechanosense the material. Some transplants or synthetic materials need to be physically modified by electrical plasma, laser engraving, etc., to achieve the desired properties. These approaches should also be discussed in the paper [https://doi.org/10.1177/19476035221075951] [https://doi.org/10.1016/j.msec.2019.02.070]

Author Response

Reviewer 2:

Point by point response to reviewer 2

The authors responded satisfactorily to all my comments and made the necessary changes to the manuscript.

Response: The authors are happy to hear that the first round corrections meet the reviewer’s expectations.

However, after reading the revised manuscript, a few questions came to my mind.

  1. Mechanical properties of scaffolds define the physiological relevance of formed implant. Some bioengineering strategies can be applied to achieve the physiological relevance of the implant. I recommend discussing some novel ideas in this area [https://doi.org/10.3390/biomedicines11030745] [https://doi.org/10.1016/j.compbiomed.2023.106804].

Response: The authors thank the reviewer for the novel literature recommended providing innovative impulses. Hence, they integrated and discussed it now in the manuscript. The very novel publication of Klabukov et al., 2023 describes a completely other tissue to be reconstructed (bile duct). Nevertheless, we refer to it now in 2.8 to underline that a radial gradient might also be interesting to mimic the ACL hierarchical structure (with epi-, peri-, endoligaments) and in 2.8.2 to hint at the degradation profile influenced by the com-/position of the layers.

The publication of Rafieyan et al., 2023 was integrated in section 2.7.

  1. Another part that should be discussed in the paper is cellular compatibility, i.e. the ability of cells to seed and mechanosense the material. Some transplants or synthetic materials need to be physically modified by electrical plasma, laser engraving, etc., to achieve the desired properties. These approaches should also be discussed in the paper [https://doi.org/10.1177/19476035221075951] [https://doi.org/10.1016/j.msec.2019.02.070]

Response: The approach of the study of Baranovski et al., 2022 was critically discussed in relation to a decellularized enthesis graft in 2.8.1. treated with laser perforation.

Since we did not talk before about collagen as an important component of the ACL which can be chemically adapted to mimic the enthesis histoarchitecture we cite now the review of Liu et al., 2019 and our own work using this functionalization strategy.
